# Lipids as Emerging Biomarkers in Neurodegenerative Diseases

**DOI:** 10.3390/ijms25010131

**Published:** 2023-12-21

**Authors:** Justin Wei, Li Chin Wong, Sebastian Boland

**Affiliations:** Prevail Therapeutics, a Wholly Owned Subsidiary of Eli Lilly and Company, New York, NY 10016, USA; justin.wei@lilly.com (J.W.); wong_li_chin@lilly.com (L.C.W.)

**Keywords:** lipids, biomarkers, neurodegenerative diseases, bis(monoacylglycero)phosphate (BMP), gangliosides, sphingolipids, glycosphingolipids, progranulin, frontotemporal dementia, Parkinson’s disease, Gaucher disease, Alzheimer’s disease, amyotrophic lateral sclerosis, multiple sclerosis, Tay–Sachs disease, Niemann–Pick disease

## Abstract

Biomarkers are molecules that can be used to observe changes in an individual’s biochemical or medical status and provide information to aid diagnosis or treatment decisions. Dysregulation in lipid metabolism in the brain is a major risk factor for many neurodegenerative disorders, including frontotemporal dementia, Alzheimer’s disease, Parkinson’s disease, and amyotrophic lateral sclerosis. Thus, there is a growing interest in using lipids as biomarkers in neurodegenerative diseases, with the anionic phospholipid bis(monoacylglycerol)phosphate and (glyco-)sphingolipids being the most promising lipid classes thus far. In this review, we provide a general overview of lipid biology, provide examples of abnormal lysosomal lipid metabolism in neurodegenerative diseases, and discuss how these insights might offer novel and promising opportunities in biomarker development and therapeutic discovery. Finally, we discuss the challenges and opportunities of lipid biomarkers and biomarker panels in diagnosis, prognosis, and/or treatment response in the clinic.

## 1. Introduction

Lipids are a structurally diverse class of organic molecules that are essential components of cell membranes and participate in multiple functions, including but not limited to energy storage and signaling [1]. Intriguingly, more than half of the human brain mass is composed of lipids [2]. In the nervous system, lipids are essential for several key functions such as synaptogenesis, neuritogenesis, and axonal insulation, and hence, it is not surprising that abnormal lipid metabolism is associated with brain pathology and neurodegeneration [3]. For these reasons, lipids are considered attractive as potential biomarkers. Biomarkers are molecules that can be used to test disease-related pathological changes in a biochemical or medical status and are present in biofluids such as blood, urine, and cerebrospinal fluid (CSF). For clinical trials, biomarkers are of utmost importance because they are needed to improve effective diagnosis and to monitor the efficacy of interventional therapies. Thus, considering an unmet need for novel biomarkers, lipids are taking center stage.

The lipidome refers to the global profile of lipid classes and their corresponding lipid species. Lipids can be broadly categorized as glycerophospholipids (GPLs), neutral lipids (NLs), and sphingolipids (SLs) (Figure 1A). The latter are extremely diverse in nature and predicted to encompass tens of thousands of species [4], and an imbalance in SL levels is associated with metabolic disease processes. Certain ceramides have recently been identified to be good predictors of cardiovascular mortality, enabling improved risk assessment of cardiovascular disease compared with conventional clinical markers [5,6,7,8]. Such advances in lipid biomarker discoveries are in large part due to advances in modern research techniques allowing large-scale determination of hundreds to thousands of individual lipid species that make up the lipidome.

In this short review, we will summarize key knowledge in lipid metabolism in neurodegenerative diseases (NDs). We give examples of recent advances in abnormal lysosomal lipid metabolism focusing on the unique anionic GPL bis(monoacylglycerol)phosphate (BMP), SLs, and glycosphingolipids (GSLs) in the most common NDs such as frontotemporal dementia (FTD), Parkinson’s disease (PD), Gaucher disease (GD), Alzheimer’s disease (AD), and amyotrophic lateral sclerosis (ALS, also known as Lou Gehrig’s disease), multiple sclerosis (MS), Tay–Sachs disease (TSD), and Niemann–Pick disease (NP). Finally, we discuss the opportunities and challenges of utilizing lipid biomarkers or lipid biomarker panels in diagnosis, prognosis, or treatment response.

## 2. Lipids in Neurodegenerative Diseases

### 2.1. The Fundamentals of Lipids and Lipidomic Analysis

Lipids are crucial for the function and integrity of cells. Lipids are highly diverse in nature and have multiple functions in the cell: they serve as (i) energy storage, (ii) essential constituents of membranes that provide compartmentalization within the cell and separate cells from their surroundings [9], (iii) signaling molecules [10], (iv) posttranslational modifications (i.e., by increasing binding affinity to membranes, affecting folding and stability, and regulating association with other proteins) [11], and (v) constituents in membrane microdomains known as “lipid rafts” [12]. The bulk of cellular (membrane) lipids are classified into GPLs, SLs, and NLs such as triglycerides (TGs) and sterols (with cholesterol (CHOL) being the main sterol in mammalian cells) (Figure 1A). GPLs are further divided into subclasses including phosphatidic acid (PA), phosphatidylserine (PS), phosphatidylethanolamine (PE), phosphatidylcholine (PC), and phosphatidylglycerol (PG) based on variations in their polar headgroup (Figure 1A, Glycerophospholipids). Bis(monoacylglycero)phosphate (BMP) (also known as lysobisphosphatidic acid or LBPA) is a unique GPL in that the stereochemical configuration of BMP differs from all other GPLs with the phosphodiester moiety linked to positions *sn-1* and *sn-1*′ of glycerol rather than to positions *sn-1* and *sn-3* (Figure 1A, Glycerophospholipids, red “*sn-1*”) [13,14]. BMP is a minor lipid constituent of cells highly enriched in the endosomal–lysosomal system.

SLs were first characterized by J. L. W. Thudichum in brain extracts in 1884 and named after the mythological sphinx because of their enigmatic nature [15]. SLs constitute the most structurally diverse lipid class and play important roles in signal transduction and cell recognition, and disorders of sphingolipid metabolism have a particular impact on nervous tissue [15]. This diversity arises from combinations of the structural components in the ceramide backbone, with dozens of possible long-chain base (LCB) and fatty acid (FA) residues and the addition of hundreds of possible head groups attached to the carbon-1 hydroxyl group (Figure 1A, Sphingolipids, red “1” above the carbon) [4,16]. Phosphorylation of the central SL ceramide leads to formation of the phosphosphingolipid ceramide 1-phosphate (C1P) and addition of the phosphobases phosphocholine or phosphoethanolamine to the formation of either sphingomyelin (SM) or of the SM analogue ceramide phosphoethanolamine (which is only produced in minute amounts in mammalian cells but is a principal membrane sphingolipid in invertebrates such as Drosophila). GSLs are formed by the sequential addition of multiple sugar moieties to SLs. For instance, gangliosides (GG), a subclass of GSLs, possess a variable number of negatively charged sialic acid residues (Figure 1A). It is estimated that the sphingolipidome is composed of thousands of individual distinct structures [16], however, understanding the biological relevance of this diversity remains challenging [17].

Omic-based platforms (lipidomics, proteomics, and metabolomics) are powerful tools to identify potential underlying disease mechanisms and pathways as well as disease biomarkers. Among these platforms, lipidomics is the fastest growing technique with an average increase in publication output by over 20% per year (since 2017) according to a “lipidomics” key word search in PubMed (Figure 1B, upper panel). The total number of publications per year that include lipidomics as a key word has increased from just over a hundred results in 2010 to almost two thousand results in 2021 (Figure 1B, lower panel). This impressive increase in scientific output is in large part due to technological advances in mass-spectrometry-based lipid analysis.

Lipidomic technologies have enhanced our knowledge about lipid functions even at the level of individual species [18]. Lipidomic techniques can loosely be divided into two approaches: shotgun lipidomics utilizing a direct infusion of a sample and liquid-chromatography-based lipidomics, both utilizing a mass spectrometer [19]. Irrespective of the analytical technique, care must be taken with interpreting the resulting data. For example, while tremendous progress with software-assisted assignments of lipid species has been made [20,21,22,23,24,25], it is essential to confirm the lipid species assignments manually to avoid false annotation [26]. Steps to ensure data integrity include but are not limited to (i) comparing the lipid fragments to databases, (ii) correlating the retention times of identified species with available lipid standards (in the case of a liquid-chromatography–mass-spectrometry-based data acquisition), (iii) matching the elemental composition of the identified lipids with the accurate masses of the precursor ions, and (iv) inspecting the molecular adducts of parental ions detected in the most abundant form. Finally, it should be noted that the data acquisition can differ in the analytical coverage (“targeted” vs. “untargeted” analysis). In the last few years, recommendations for good practice in lipidomics have been published [27,28,29,30]. In addition, a checklist summarizing key details of lipidomic analyses with the aim to harmonize the field has just been published [31].

To analyze lipids by mass spectrometry, lipids are extracted with organic solvents from cells, fluids, or tissues. Because of their highly diverse nature, the choice of the extraction procedure as well as the addition of appropriate internal standards is critical to quantitative lipidomics [32]. Chloroform/methanol-based two-phase lipid extraction methods according to Folch [33] or Bligh and Dyer [34] have been most commonly used, but multiple methods to extract lipids are available and many of them serve different objectives: (i) rapid one-step extraction procedures enabling high-throughput analysis of lipids [35], (ii) replacement of toxic chloroform with less toxic solvents to improve health and environmental safety [35,36,37], (iii) recovery of specific lipids (i.e., acidic lipids present in trace amounts, such as PA and LPA) [38]. Discussions of different extraction methods and their use depending on biological matrices can be found, for example, in this recent publication [39].

### 2.2. Lysosomal Lipid Catabolism in Health and Disease

More than half of the human brain mass is composed of lipids [2], but little is known about brain lipid metabolism in health and disease. Given their essential roles in cellular function and integrity, it is not surprising that abnormal lipid metabolism, and in particular abnormalities in the catabolism of GSLs/GGs which are especially abundant in the central nervous system, is associated with neurodegeneration and neuroinflammation [5,40,41].

Lysosomes are significant sites for catabolism of lipids [42]. Under normal circumstances (the internal environment of lysosomes is acidic, with a pH range of 4.6–5), cationic lysosomal lipid-binding proteins and respective hydrolases are electrostatically attracted to the negatively charged luminal surface of BMP-rich ILVs [43]. BMP serves as a critical constituent of the limiting membrane and the luminal leaflet of intraluminal vesicles (ILVs) and provides a platform for adequate lipid catabolism [44,45]. In the case of GSLs/GGs the end standing sugar moieties are removed by acid glycosidases/hydrolases in a sequential manner and might require the assistance of “helper” proteins such as saposins or GM2-activator protein (Figure 2A) to degrade lipids into their corresponding components: sugars, FAs, and LCBs. Most lipid degradation products are subsequently recycled by the cell to generate new lipids.

Lysosomal storage disorders (LSDs) lead to neurodegeneration and consequently premature death. In LSDs, degradation of (lipid) substrates is impaired, and it has been suggested that cells may compensate by elevation of BMP as tested in cultured fibroblasts and plasma samples from patients with LSDs [46]. Hence, BMP may serve as a useful biomarker for a subset of LSDs.

Much of the described lipid degradation model has been intensely discussed in several other publications [42,47,48]. The positively charged amino acids lysine, arginine, and histidine on the surface of lysosomal proteins (Table 1) play a key role in the lipid–protein interaction. Interestingly, histidine is the only amino acid with a titratable basic side chain (pka ≈ 6) within the physiological range. At an acidic pH (lysosomal pH ≈ 4.5–5.0 [45]), histidine residues in enzymes and/or helper proteins such as β-glucosylceramidase (GCase) and GM2-AP are positively charged, which may result in enhanced electrostatic interaction with BMP ensuring efficient lipid–protein interactions.

**Figure 2 ijms-25-00131-f002:**
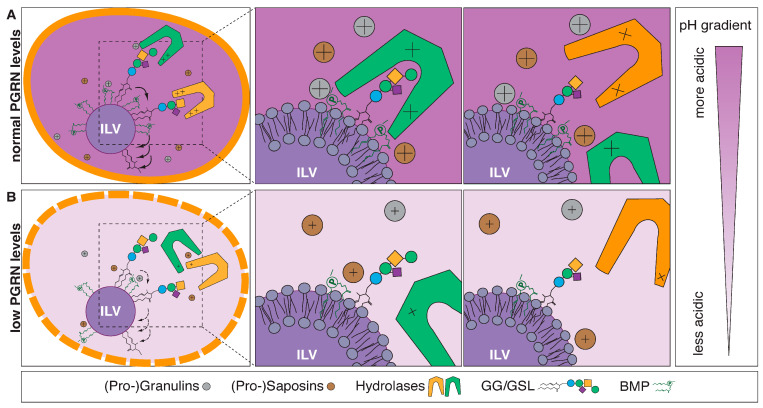
Model of lysosomal lipid degradation. (**A**) Under normal circumstances, cationic lysosomal lipid-binding proteins and respective hydrolases are electrostatically attracted to the negatively charged luminal surface of BMP-rich ILVs. BMP serves as a critical constituent of the limiting membrane and the luminal leaflet of intraluminal vesicles (ILVs) and provides a platform for adequate, sequential (indicated by arrows) lipid catabolism [44,45]. The end standing sugar moieties of GSLs/GGs are removed by acid glycosidases/hydrolases in a sequential manner and might require the assistance of “helper” proteins such as saposins or GM2-activator protein. (**B**) PGRN/granulin deficiency leads to reduced BMP levels through a yet to be discovered mechanism. Reduced BMP levels contribute to impaired hydrolase activities and drive accumulation of lipid substrates such as GlcSph and GGs. Eventually, this likely leads to lysosomal dysfunction and downstream consequences, including neuroinflammation and neurodegeneration. Adopted from Boland et al. [44] and Simon et al. [49].

*Frontotemporal dementia-GRN* (FTLD-*GRN*) is one of the most common dementias in individuals under the age of 60 and is caused by progressive nerve cell loss in the frontal and/or temporal lobes [50]. Mutations in the granulin precursor (*GRN*) gene encoding progranulin (PGRN) are a major cause of familial FTD [51,52]. Recently, two independent studies demonstrated that deficiency in PGRN and/or granulins leads to decreased levels of the endosomal–lysosomal lipid BMP [53,54]. The seminal study by Logan et al. suggests that PGRN and/or GRN peptides stabilize BMP, which in turn stimulates the activity of the lysosomal enzyme GCase. Thus, deficiency of PGRN and/or GRN peptides leads to a reduction in BMP levels, thereby lowering GCase activity resulting in an accumulation of the shunt metabolite glucosylsphingosine (GlcSph) (Figure 2B). The reduction in BMP levels and the accumulation of GlcSph were evident in isolated liver and brain tissues from *Grn*^−/−^ mice compared to age-matched *Grn*^+/+^ mice. Similarly, BMP levels trended lower in CSF isolated from *GRN* mutation carriers, whereas GlcSph levels were increased in plasma samples from *GRN* mutation carriers. Remarkably, while no differences in BMP levels were observed between asymptomatic and symptomatic subjects, GlcSph levels were increased in symptomatic patients only [53]. Likewise, in a second independent study, Boland et al. also discovered reduced levels of BMP in brain tissues from *Grn*-deficient mice and in HeLa cell lines with complete loss of PGRN. Importantly, this study demonstrated reduction of BMP (22:6/22:6) species in frontal and occipital lobes from postmortem human tissues with *GRN*-mutation-related FTD. The authors hypothesized that PGRN and/or GRN peptides facilitate GG catabolism by maintaining BMP levels and that PGRN and/or GRN deficiency in lysosomes leads to gangliosidosis (Figure 2B). Indeed, loss of PGRN results in increased levels of GGs that likely contribute to neuroinflammation and neurodegeneration. Additionally, the authors performed experiments in *Grn*^−/−^ cells with PGRN expression restored via lentiviral infection with untagged human PGRN cDNA resulting in normalization of BMP and GG levels. Likewise, the authors showed that BMP supplementation to cells in culture can rescue the GG phenotype in *Grn*^−/−^ cells [45]. Hence, PGRN is a key regulator of GG and GSL degradation due to its ability to maintain lysosomal levels of BMP that can modify the activity of various hydrolytic enzymes including GCase in the lysosomes (Figure 2). This unexpected lysosomal biology of PGRN and/or granulins, respectively, offers promising opportunities in lipid biomarker development such as BMP and GGs in FTD. A recent study demonstrated that rAAV delivery of individual GRNs can partially rescue inflammation and lysosomal dysfunction (including dysregulated lipid metabolism, most notably BMP and GG metabolism) in a mouse model with PGRN deficiency [54].

**Table 1 ijms-25-00131-t001:** Lysosomal proteins and their positively charged residues.

Lipid-Binding Proteins	References	Positively Charged Residues
Acid ceramidase	[55]	Arginine—16; Lysine—26; Histidine—5
β-galactosidase	[56]	Arginine—28; Lysine—30; Histidine—19
β-glucosylceramidase	[57]	Arginine—23; Lysine—23; Histidine—18
Galactosylceramidase	[58]	Arginine—25; Lysine—34; Histidine—16
GM2 activator protein	[56]	Arginine—4; Lysine—11; Histidine—3
Heat shock protein 70 A	[59]	Arginine—25; Lysine—34; Histidine—16
Progranulin	[53]	Arginine—33; Lysine—16; Histidine—27
Granulin A		Arginine—1; Lysine—2; Histidine—2
Granulin B		Arginine—2; Lysine—2; Histidine—2
Granulin C		Arginine—0; Lysine—0; Histidine—2
Granulin D		Arginine—2; Lysine—1; Histidine—3
Granulin E		Arginine—7; Lysine—1; Histidine—3
Granulin F		Arginine—2; Lysine—0; Histidine—3
Granulin G		Arginine—2; Lysine—0; Histidine—5
Paragranulin		Arginine—2; Lysine—0; Histidine—0
Prosaposin	[56]	Arginine—8; Lysine—43; Histidine—11
Saposin A	[60]	Arginine—1; Lysine—6; Histidine—0
Saposin B	[56,60]	Arginine—2; Lysine—3; Histidine—2
Saposin C	[56,60]	Arginine—0; Lysine—7; Histidine—1
Saposin D	[60]	Arginine—1; Lysine—7; Histidine—1

*Parkinson’s disease* (PD) is a progressive ND caused by the death of dopaminergic neurons in the substantia nigra pars compacta. It is believed that aggregation of α-synuclein is a key event in PD [61]. *GBA1* mutations are the most commonly known genetic risk factor for PD and about 10% of *GBA1* mutation carriers will develop PD [62]. *GBA1*-associated PD (*GBA*-PD) is similar to idiopathic PD, however, subjects with *GBA1* mutations display an earlier onset and higher prevalence of cognitive changes [63,64]. As mentioned earlier, *GBA1* encodes GCase, an enzyme residing in lysosomes, where it degrades its substrates glucosylceramide (GlcCer) or GlcSph. As a result of reduced GCase activity, it is suggested that GlcCer species accumulate in lysosomes and directly interact with α-synuclein. That in turn may stabilizes soluble oligomeric α-synuclein intermediates, leading to fibril formation. Inversely, α-synuclein aggregates may reduce GCase activity, creating a bidirectional pathogenic loop [65]. However, evidence that GlcCer or GlcSph accumulate in human PD or GBA-PD brains or CSF is inconclusive, and hence the disease mechanism remains to be fully elucidated [66]. Interestingly, recent work has implicated systemic alterations of the GGs GM1 and GD1a in PD tissues, as well as in non-neuronal cells (PBMCs) and neurons [67,68].

Mutations in the leucine-rich repeat kinase 2 (LRRK2) gene are among the most common risk factors in PD with the most common pathogenic variant LRRK2 G2019S accounting for about 5% of familial PD cases [69]. Studies have thoroughly established that LRRK2 activity modulates levels of BMP (22:6/22:6) in urine, in LRRK2-knockout mice, and in non-human primates treated with LRRK2 kinase inhibitors [70]. Conversely, individuals with a PD-causing LRRK2 gain-of-function mutation displayed elevated BMP (22:6/22:6) levels compared to asymptomatic mutation carriers, which were predictive of impaired cognitive performance [71]. Thus, BMP might be a useful biomarker for clinical trials of LRRK2-targeted therapies. One such example is the use of BMP in the recently completed phase 1 trial (NCT04551534) with the LRRK2 inhibitor DNL201 [72].

*Gaucher disease* (GD) is one of the most common LSDs, with a prevalence of 0.70 to 1.75 per 100,000 people, and is caused by biallelic recessive mutations in the *GBA1* gene [73]. The most common form is the non-neuronopathic GD type 1, which is distinguished by the lack of neurological manifestations as opposed to the other two forms of neuronopathic GD (nGD, GD type 2 and type 3). GD is characterized by the accumulation of GlcCer and its deacetylated toxic form GlcSph due to reduced activity of GCase [74].

Current treatments for the various types of GD are enzyme replacement therapies (ERTs), however, these treatments are often expensive and require life-long dosing [75]. Small-molecule therapies such as miglustat treatment prevent the synthesis of GlcCer and have been studied and approved for the treatment of GD (Table 1). Clinical trials with AAV-based gene therapies are ongoing (Table 1). These trials all use glycolipids as outcome measures, showcasing the use of lipids as biomarkers to measure the effectiveness of potential treatments.

*Alzheimer’s disease* (AD) is the most common type of dementia and is the cause of 60–70% of dementia cases with currently ~6 million Americans living with AD. It is predicted that this number will rise to over 14 million people by 2060 in the United States alone [76]. AD pathogenesis is characterized by the deposition of β-amyloid (Aβ) peptides [77]. Aβ peptides have been shown to bind to GGs, especially GM1, leading to alterations in the secondary structure of Aβ. This specific form of Aβ bound to GM1, known as GAβ, is found in brains exhibiting early pathological changes associated with AD and it is suggested that it may contribute to the formation of Aβ aggregates [78,79,80,81]. A further increase in the Aβ density leads to the formation of amyloid fibrils that trigger apoptosis and ultimately neurodegeneration [82].

Translational studies utilizing lipidomic approaches demonstrated ~50% increased levels of SM in CSF of prodromal AD patients compared to cognitively normal controls. Increases in significantly changed SM species ranged from ~30% for SM (d18:1/20:0) and up to ~80% for SM (d18:1/24:1). Surprisingly, no significant change in total SM was observed in the CSF of patients with mild or moderate AD [83,84]. While CSF collection is routine, it still is an inconvenient process requiring a lumbar puncture. Therefore, plasma lipids are being examined as alternative biomarkers for AD diagnosis in the future. It has been found that elevated levels of low-density lipoprotein chol and total chol in plasma were observed in AD pathologies [85]. Elevated chol levels have been implicated in Aβ aggregate formation, as chol has been shown to mediate Aβ metabolism. Animal models with increased chol levels exhibit increased Aβ levels in the brain [86]. SLs have also been suggested to regulate amyloid precursor protein (APP) by modulating γ-secretase, a transmembrane protein that contributes to the formation of Aβ peptides [87]. FAs play an important role in neuronal membrane fluidity, and it has been found that AD patients show much lower membrane fluidity than healthy controls. This lower membrane fluidity has been associated with abnormal APP metabolism and dementia [88]. While these and other lipid species have been found to be altered within AD pathologies, more work must be carried out to be able to utilize these lipids as reliable biomarkers.

It has been reported that abnormally high lysosomal pH levels can be correlated with NDs. In presenilin 1 (PS1)-knockout cell lines and mouse models, it was measured that lysosomal pH was above 5, and reduced enzyme activity was recorded compared to WT cells. PS1 is an integral membrane protein that, with other proteins in a complex, cleaves amyloid precursor protein (APP). Mutations in PS1 are the most common factor in early-onset familial Alzheimer’s disease. While increased lysosomal pH has been implicated in reduced proteolysis in lysosomes, abnormal pH levels can also negatively affect lipid catabolism as well [89] as discussed earlier.

Apolipoprotein E (ApoE) is an important protein involved with the metabolism and regulation of lipids and chol. There are three common variants of ApoE with the ApoE4 variant implicated as a genetic risk factor for AD. Chol is an important part of the myelination of nerve cells, and it was recently shown that brains of humans with the ApoE4 allele exhibited abnormal chol deposition, resulting in reduced myelination [90]. A recent study demonstrated altered lipid composition in pluripotent-stem-cell-derived APOE4 astrocytes by accumulating unsaturated TAGs in lipid droplets to a greater extent than isogenic APOE3 counterparts. Interestingly, supplementation with choline promoting phospholipid synthesis restores lipid imbalances in these cell lines [91].

*Amyotrophic lateral sclerosis* (ALS) is a progressive ND mainly affecting motor neurons. In *wobbler* mice, a partial loss-of-function mutation (L967Q) in the Vps54-encoded subunit of the Golgi-associated retrograde protein complex is responsible for motor neuron loss with features similar to ALS. Accumulation of cytotoxic sphingoid bases in isolated murine embryonic fibroblasts and spinal cords from the *wobbler* mouse line has been demonstrated. Remarkably, chronic treatment of *wobbler* mice with myriocin, an inhibitor of the first step in the de novo sphingolipid biosynthesis, significantly improved overall wellbeing and survival [92]. Like the *wobbler* mouse study, motor neurons in the spinal cord in ALS patients display elevated levels of the SLs ceramide, GlcCer, lactosylceramide, galactosylceramide, and globotriaosylceramide as well as the GGs GM3 and GM1. In addition, marked alteration in the plasma SL profile of ALS patients has been reported [93,94]. While the pathological relevance of altered lipid metabolism remains to be established, levels of SM and long-chain TG species in the CSF of ALS patients correlated with disease progression and might act as biomarkers of the disease [95,96].

*Multiple sclerosis* (MS) is an ND characterized by the demyelination and degeneration of neurons in the central nervous system. T-cells that cross a damaged blood–brain barrier recognize myelin as foreign and trigger an immune response, damaging this important sheath. Further attacks and insufficient repairs of myelin lead to the formation of lesions, or plaques. Lipids are becoming a focus of MS research as they are a major component of myelin. It has been shown that exosomes containing ceramides are released from stressed oligodendrocytes, and these exosomes may act as apoptosis signals and promote autoimmune demyelination. Increased levels of ceramides have been observed in plasma, CSF, and plaques [97]. Chol and oxysterol levels are also altered in MS pathologies [98].

*Tay–Sachs disease* (TSD), also known as GM2 gangliosidosis, is an LSD caused by mutation in the *HEXA* or *HEXB* genes. These mutations cause reduced or absent activity of the enzyme hexosaminidase A, which metabolizes the GG GM2. This leads to the accumulation of GM2 in the lysosomes, which causes neuronal loss. The fatal disease usually presents in infants or toddlers, and to date there is no approved treatment targeting the root cause of this disease. GM1, GM2, GM3, and lyso-GM2 have been proposed as biomarkers that can be used to track disease progression and potential treatments. GM1, GM2, and lyso-GM2 levels in plasma were found to be greatly elevated in Tay–Sachs patients, while GM3 levels were found to be reduced [99,100].

*Niemann–Pick disease* (NP) is an LSD that is categorized into two variants: type A/B, which is caused by a mutation in the *SMPD1* gene that leads to reduced activity of the enzyme acid sphingomyelinase (ASM), and type C, which is caused by mutations in the *NPC1* and *NPC2* genes that affect proteins involved in the transport of chol and other lipids. Type A/B presents in infants and toddlers, and most do not survive past a few years after diagnosis. Type C can present in infants, children, or adults, often fatally for infants and children. The deficiency of ASM that characterizes type A/B causes a buildup of sphingomyelin in the lysosome, which is cleaved into phosphocholine and ceramide by the enzyme. Enlarged, lipid-filled foam cells are present in the organs of type A/B patients, and neuronal loss and demyelination have been observed in the brains of patients. In type C Niemann–Pick disease, elevated chol, BMP, GlcCer, LacCer, and GM3 levels were observed in the spleen and liver. In the brain, elevated GSL levels were reported along with tau protein filaments and Aβ proteins [101,102].

As apparent by the above examples, changes in the lipidome and particularly in BMP and GSLs appear to be common features of several NDs. Thus, it is not surprising that lipids and their intermediates gain attention as potential biomarkers of these disorders and may serve as diagnostic tools in the clinic.

### 2.3. Lipidomic-Based Biomarkers in Clinical Trials

It is much anticipated that lipids might fill missing gaps as biomarkers in the clinic, as they play central roles in disease metabolism and thus represent a direct readout of the studied disease phenotypes. Currently, there are dozens of clinical trials exploring the utility of lipid biomarkers for various disease indications. A few examples with respect to NDs are given in Table 2 below:

A potential therapy for PD-GBA involves the delivery of a healthy copy of the *GBA1* gene using an AAV9 vector to restore normal GCase protein and activity levels. LY3884961 is in phase 1 clinical trials to determine the efficacy of such a therapy, with the potential to also be used in the treatment of GD type 1 and type 2. GSLs such as GlcCer and GlcSph will be measured in this clinical trial to determine the effectiveness of the therapy (NCT04127578).

In the LEAP trial (NCT02843035), venglustat is being investigated for the treatment of GD type 3. Venglustat is a glucosylceramidase synthase inhibitor that is able to cross the blood–brain barrier, a property that conventional therapies lack. A primary endpoint of the trial is the change in concentration of the lipids GlcCer and GlcSph in CSF [103], a first for such a trial, as lipid biomarkers have usually been regarded as a secondary endpoint or exploratory measurement.

Miglustat, a treatment of GD type 1, is also being investigated for the treatment of other NDs such as Tay–Sachs disease. Miglustat is a competitive and reversable inhibitor of the enzyme glucosylceramide synthase, which executes the first step of GSL synthesis. Therefore, the GG GM2 is used to evaluate the effectiveness of miglustat treatment in NDs caused by GM2 lipid accumulation in lysosomes (NCT00672022).

As an ERT for Niemann–Pick disease type A, olipudase alfa was found to reduce LDL chol by ~35% on average and TG by ~50% on average. HDL chol increased by ~107% on average. Sphingomyelin levels decreased by ~24% on average and lyso-sphingomyelin levels decreased by ~87% on average. These results are an example of a completed clinical trial where lipids were used as secondary endpoints to determine the efficacy and safety of an investigational treatment (NCT02292654).

In these trials, lipidomic analysis is employed as a read-out of disease progression and/or drug efficacy. Further developments in lipidomics will allow for wider use of the technique, with applications including (i) the identification of subjects that are at risk of developing an ND, (ii) preventive therapy management, and (iii) patient stratification. However, as a single marker can be linked to different conditions, it is most likely that biomarker panels are needed rather than single biomarkers. Therefore, the next step is to expand from single lipid markers to lipid panels or lipidomic readouts, enabling a more comprehensive understanding of lipid-related pathophysiologies in NDs. There are additional shortcomings with current lipidomic techniques that need to be addressed. Firstly, broad lipid panels could be affected by behaviors and factors unrelated to the disease. Genetic, physical, and nutritional factors can all affect whole body lipid levels, which could cause some bias in lipid biomarker panels [104]. Secondly, exercise has been shown to decrease lipid profiles related to cardiovascular disease [105]. Panels will need to be carefully designed to assess lipids that are directly influenced by a disease or treatment state. Thirdly, for lipidomic panels to become a routine test, a standardized approach for the reporting of lipidomics data needs to be adopted to ensure that data integrity is preserved across sample handling, sample analysis, and data processing [106] as discussed earlier. Finally, for biomarkers to gain widespread acceptance in clinical settings, assays would need to be inexpensive, non-invasive, and simple to perform without the need for complex equipment [104]. Working with regulatory agencies regarding the reporting and presentation of lipidomic data would greatly aid the broader lipidomic community in solving some of the above challenges.

### 2.4. Future Perspectives and Closing Remarks

NDs are a profound public health issue costing over USD 800 billion per year worldwide [107]. As the number of elderly citizens increases, so will the costs to society. In addition to the financial costs, there is an immense emotional burden on patients and their caregivers with direct implications on the patients’ lives [108]. Since dysregulation in lipid metabolism in the brain is a major risk factor for many NDs, lipid biomarkers can be used to advise patients and healthcare providers about treatment options and their effects on disease progression. Many novel biomarkers that reflect a broad range of pathological events involved in the progression of NDs have been reported. Here, we have provided an overview of lipid biomarker and biomarker panels in diagnosis, prognosis, or treatment response in the clinic. We discussed the challenges and opportunities of lipid biomarkers and (without claiming completeness) we provided some examples of lipid biomarkers in NDs in the previous sections. With many new lipidomics tools available to scientists, our array of biomarkers will expand, helping us improve the drug development process and patient stratification. Understanding the relationship between measurable biological processes and clinical outcomes is critical to expand our treatments for NDs. Effective biomarkers to improve diagnosis (personalized medicine: “the right treatment to the right patient, at the right dose at the right time”) will certainly aid the development of interventional therapies by establishing target engagement and finding the effective dose range. In conclusion, lipidomics-driven biomarker discovery and application have begun. Hence, it is only a matter of time before new lipid tests are routinely used in the clinic.

## Figures and Tables

**Figure 1 ijms-25-00131-f001:**
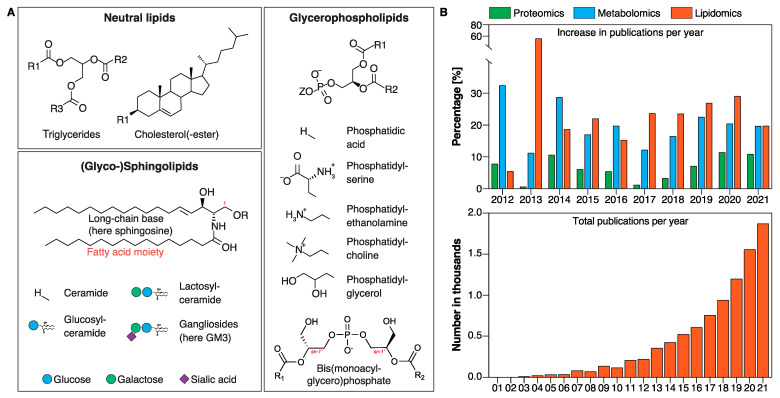
Lipid classes and their corresponding lipid species. (**A**) Lipids are broadly categorized into neutral lipids (panel **top left**), sphingolipids (panel **bottom left**), and glycerophospholipids (panel **right**). (**B**) Lipidomics is the fastest growing technique with an average increase in publication output by over 20% per year (since 2017) according to a “lipidomics” key word search in PubMed.

**Table 2 ijms-25-00131-t002:** Biomarkers measured in ND clinical trials.

Biomarker(s)	Treatment	Proposed Activity	Disease	Clinical Trial
Lyso-GlcCer	Venglustat	Glucosylceramide synthase inhibitor	GD type 3	NCT02843035
GlcSph	Velaglucerase alfa	ERT of glucocerebrosidase	GD type 1	NCT05529992
GSLs	PR001/ LY3884961	AAV carrying GBA1 to reverse the deficiency of β-glucocerebrosidase	GD type 1	NCT05487599
GSLs	PR001/ LY3884961	AAV carrying GBA1 to reverse the deficiency of β-glucocerebrosidase	GD type 2	NCT04411654
GSLs	PR001/ LY3884961	AAV carrying GBA1 to reverse the deficiency of β-glucocerebrosidase	PD	NCT04127578
BMP	DNL201	LRRK2 kinase inhibitor	PD	NCT04551534
SM Chol TG	Olipudase alfa	ERT of acid sphingomyelinase	NP type A/B	NCT02292654
FAs TG	N/A (observational)	N/A	AD	NCT03070535
FAs	Choline	Stabilizing lipid metabolism	AD	NCT05880849
GM2	Miglustat	Inhibit the formation of GM2 ganglioside	TSD	NCT00672022
GM2	N/A (observational)	N/A	TSD	NCT01869270
Peroxidized lipids	Deferiprone	Reducing oxidative stress	ALS PD	NCT02880033
FAs Chol	N/A (observational)	N/A	ALS	NCT02572479

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
