# Peer review of "Lipids as Emerging Biomarkers in Neurodegenerative Diseases"

_ijms, 2023, doi:10.3390/ijms25010131_

Round 1

Reviewer 1 Report

Comments and Suggestions for Authors

This review focuses on the usefulness of lipidome analysis in the search for a panel of biomarkers for the diagnosis of some neurodegenerative diseases. Finally, the importance of some specific lipids in these diseases and some therapeutic approaches are discussed.

The article is very well written, and the information is clearly presented. The authors present their arguments clearly. Current relevant information is presented and discussed. However, in my opinion, it would be interesting and usefull if the authors consider and explained some aspects.

1- Lines 34 and 36 and in Figure 1, panel A. Check the order of classification of lipids. It should be consistent.

2- The molecular mechanisms involving lipids in frontotemporal dementia could be better understood if a figure showing the main aspects involved was included (This figure could also be extended to other pathologies).

3- In lines 227-228, very old references are given. Are these studies that have been continued or discontinued? Given the recent controversy about Ab peptides in AD, it is suggested that this information be updated or deleted.

4- Lines 236-237. Although other changes in lipid metabolism and composition in AD are discussed in detail in other reviews, it would be very useful to include a very brief summary of conclusions here (about 3 to 6 main aspects or so).

5- It would be useful to list “multiple sclerosis” among the CNS pathologies presented. There is important information about lipids involved in this disease.

6- What about Parkinson´s disease? There is no research on lipidomics in PD?

7- What criteria were used to select the information contained in Table 1? There seems to be a lack of information on AD and ALS. On the other hand, this table includes lipid biomarkers for pathologies not included in the text, such as Parkinson´s disease, Tay-Sachs disease, and Fabry´ syndrome. It would be useful to align the information between the table and the text and present the entire group of included diseases in both.

Author Response

We thank the reviewer for his/her time and are pleased by the reviewer’s positive feedback. We are grateful for the reviewer’s useful insights and suggestions. A point-by-point response is provided below. 

1- Lines 34 and 36 and in Figure 1, panel A. Check the order of classification of lipids. It should be consistent.
> The order of the classification of lipids has been changed to make the text and the Fig. 1A consistent.  

2- The molecular mechanisms involving lipids in frontotemporal dementia could be better understood if a figure showing the main aspects involved was included (This figure could also be extended to other pathologies).
> As suggested we have now added extra text and a figure that explains the underlying pathological lysosomal lipid biology with focus on FTD-GRN. 

3- In lines 227-228, very old references are given. Are these studies that have been continued or discontinued? Given the recent controversy about Ab peptides in AD, it is suggested that this information be updated or deleted.
> The authors agree with the fact that these are old references and some of these studies have been discontinued. For these reasons, this passage has now been deleted. 

4- Lines 236-237. Although other changes in lipid metabolism and composition in AD are discussed in detail in other reviews, it would be very useful to include a very brief summary of conclusions here (about 3 to 6 main aspects or so).
> The AD paragraph has been extended to capture the most discussed conclusions in lipid metabolism regarding AD. 

5- It would be useful to list “multiple sclerosis” among the CNS pathologies presented. There is important information about lipids involved in this disease.
> A paragraph on lipids involved in the pathology of multiple sclerosis has been added. 

6- What about Parkinson´s disease? There is no research on lipidomics in PD?
> Although dysregulation in lipid metabolism in Parkinson’s Disease was previously included, we have now expanded on the subject. Additionally, to increase visibility, we have now italicized the different diseases in this section. 

7- What criteria were used to select the information contained in Table 1? There seems to be a lack of information on AD and ALS. On the other hand, this table includes lipid biomarkers for pathologies not included in the text, such as Parkinson´s disease, Tay-Sachs disease, and Fabry´ syndrome. It would be useful to align the information between the table and the text and present the entire group of included diseases in both.
> We thank the reviewer for pointing out our shortcomings with information on AD and ALS. We have added more clinical trials for these diseases. In addition, we have edited the text to align with the table and added information on PD, Tay-Sachs, and Fabry.

Reviewer 2 Report

Comments and Suggestions for Authors

This short review summaries the use of lipids as biomarkers for neurodegenerative diseases. After a brief introduction to lipid structures, examples for neurodegenerative diseases with alterations in lipids are presented. The manuscript is clearly written and well structured aiming at readers not familiar with the field.

Personally, I would have wished to get further information on technical aspects as probe sampling and analysis techniques evaluating the feasibility of lipidomics as a diagnostic tool.

Author Response

Personally, I would have wished to get further information on technical aspects as probe sampling and analysis techniques evaluating the feasibility of lipidomics as a diagnostic tool.
> We thank the reviewer for his/her time and are pleased by the reviewer’s positive feedback. As suggested, we have now expanded the section on the feasibility of lipidomics as a diagnostic tool (line 372ff). 

Reviewer 3 Report

Comments and Suggestions for Authors

The manuscript by Wei et, al provides an intriguing overview of the potential utility of lipids as biomarkers in the context of neurodegenerative diseases. While the central theme holds promise, there are a few aspects that require further development and elaboration to enhance the depth and impact of the review before its acceptance are as follows:

1. The review establishes the premise that dysregulation in lipid metabolism contributes significantly to neurodegenerative disorders such as frontotemporal dementia, Alzheimer's disease, Parkinson's disease, and amyotrophic lateral sclerosis. However, it would be prudent to strengthen this assertion by incorporating references to seminal studies or key findings that establish a clear and direct link between lipid metabolism dysregulation and the pathogenesis of these specific disorders. By providing concrete evidence, the review will augment its claims and lend more weight to its arguments.

2.  The manuscript addresses various lipid classes and their relevance but omits an essential lipid-related pathway, namely oxytosis/ferroptosis. Given the emerging importance of this pathway in the context of neurodegenerative diseases, it is advisable to introduce a dedicated section that elucidates the connection between oxytosis/ferroptosis and lipid metabolism dysregulation. Integrating this aspect will enrich the review's scope and demonstrate a comprehensive understanding of the topic.

3.While the manuscript briefly mentions bis(monoacylglycero)phosphate and (glyco-)sphingolipids as promising lipid classes, it falls short of providing a comprehensive understanding of their roles and significance in neurodegenerative diseases. To address this, consider expanding upon the functions of these lipid classes within cellular processes and how their dysregulation could contribute to disease progression. This expansion will ensure that readers gain a more nuanced perspective on the potential implications of these lipids as biomarkers.

4. To underscore the practical implications of using lipid biomarkers in clinical settings, it is recommended to integrate specific case studies or clinical scenarios where lipid biomarkers have been proposed or utilized. By presenting real-world applications, the review will bridge the gap between theoretical insights into abnormal lysosomal lipid metabolism and their translation into actionable diagnostic or therapeutic strategies.

5. Exploration of Challenges and Future Directions:

While the manuscript briefly suggests to challenges associated with lipid biomarker panels, further exploration of these challenges, along with potential strategies to overcome them, will enhance the review's completeness. Discussing the obstacles that need to be addressed in order to implement lipid biomarkers effectively in clinical practice will provide readers with a more comprehensive understanding of the field's current state and its future trajectory.

In conclusion, the manuscript lays a promising foundation for a comprehensive review on the role of lipids as biomarkers in neurodegenerative diseases. However, to ensure the scholarly rigor and depth of the review, addressing the aforementioned points is essential.

Author Response

We thank the reviewer for his/her time and are pleased by the reviewer’s positive feedback. We are grateful for the reviewer’s useful insights and suggestions. A point-by-point response is provided below.

1. The review establishes the premise that dysregulation in lipid metabolism contributes significantly to neurodegenerative disorders such as frontotemporal dementia, Alzheimer's disease, Parkinson's disease, and amyotrophic lateral sclerosis. 

However, it would be prudent to strengthen this assertion by incorporating references to seminal studies or key findings that establish a clear and direct link between lipid metabolism dysregulation and the pathogenesis of these specific disorders. By providing concrete evidence, the review will augment its claims and lend more weight to its arguments.
> We have now added additional references providing evidence for a direct link between lipid metabolism dysregulation and the pathogenesis of a selection of neurodegenerative disorders. 

 2.  The manuscript addresses various lipid classes and their relevance but omits an essential lipid-related pathway, namely oxytosis/ferroptosis. Given the emerging importance of this pathway in the context of neurodegenerative diseases, it is advisable to introduce a dedicated section that elucidates the connection between oxytosis/ferroptosis and lipid metabolism dysregulation. Integrating this aspect will enrich the review's scope and demonstrate a comprehensive understanding of the topic.
> We agree with the reviewer that ferroroptosis plays an essentioal role in the context of ND, however, we believe that this is beyond the scope of this review.

3. While the manuscript briefly mentions bis(monoacylglycero)phosphate and (glyco-)sphingolipids as promising lipid classes, it falls short of providing a comprehensive understanding of their roles and significance in neurodegenerative diseases. To address this, consider expanding upon the functions of these lipid classes within cellular processes and how their dysregulation could contribute to disease progression. This expansion will ensure that readers gain a more nuanced perspective on the potential implications of these lipids as biomarkers. 
> As suggested, we have now added extra text and a figure that explains the underlying pathological lysosomal lipid biology with focus on FTD-GRN.

4. To underscore the practical implications of using lipid biomarkers in clinical settings, it is recommended to integrate specific case studies or clinical scenarios where lipid biomarkers have been proposed or utilized. By presenting real-world applications, the review will bridge the gap between theoretical insights into abnormal lysosomal lipid metabolism and their translation into actionable diagnostic or therapeutic strategies.
> As suggested, we have now added extra text that describes case studies with lipid biomarkers used to support clinical trial studies.

5. Exploration of Challenges and Future Directions: While the manuscript briefly suggests to challenges associated with lipid biomarker panels, further exploration of these challenges, along with potential strategies to overcome them, will enhance the review's completeness. Discussing the obstacles that need to be addressed in order to implement lipid biomarkers effectively in clinical practice will provide readers with a more comprehensive understanding of the field's current state and its future trajectory.
> We expanded our initial section on the feasibility of lipidomics as a diagnostic tool and its future trajectory.

Round 2

Reviewer 1 Report

Comments and Suggestions for Authors

The authors have conveniently addressed the questions and comments, consequently the manuscript has been improved in precision.